# How does the UK childcare energy-balance environment influence anthropometry of children aged 3–4 years? A cross-sectional exploration

Kathryn R Hesketh,[1,2] Sara E Benjamin-Neelon,[1,2] Esther M F van Sluijs[1]

[1]Centre for Diet and Activity Research and MRC Epidemiology Unit, University of Cambridge, Cambridge, UK
[2]Department of Health, Behavior and Society, Johns Hopkins Bloomberg School of Public Health, Baltimore, Maryland, USA

**Correspondence to**
Dr Kathryn R Hesketh; kathryn.hesketh@ucl.ac.uk

## ABSTRACT

**Objectives** To assess the association between time spent in care, the childcare energy-balance environment, and preschool-aged children's body mass index z-score (z-BMI), waist-to-height ratio (WHR) and sum of skinfold thickness (SST).

**Design** Cross-sectional study.

**Setting and participants** Children aged 3–4 years were recruited from 30 childcare centres in Cambridgeshire (UK) in 2013.

**Main outcome measures** Objectively measured height and weight was used to calculate z-BMI; waist circumference and height were used to generate WHR; subscapular and tricep skinfolds were used to calculate SST. Associations between childcare attendance, the nutrition, physical activity, and overall childcare environment, and three anthropometric outcomes were explored using two-level hierarchical regression models, adjusting for demographic and family based confounders.

**Results** Valid data were available for 196 children (49% female). Time spent in care, the nutrition, physical activity and overall childcare environment were not associated with children's z-BMI, WHR and SST.

**Conclusions** Childcare environment and level of attendance were not associated with UK preschool-aged children's anthropometry. The childcare environment has been central to intervention efforts to prevent/reduce early childhood obesity, yet other factors, including child-level, family level, wider environmental and policy-level factors warrant substantial attention when considering obesity prevention strategies for young children.

## Strengths and limitations of this study

► We used objective measures of UK preschool-aged children's anthropometric indices to provide novel information about how they are associated with childcare attendance and environment.

► This was a relatively small UK childcare-based sample, but our outcomes and exposures were normally distributed, providing sufficient heterogeneity to explore our research questions.

► We used hierarchical regression analysis to take account of study design and potential clustering at the centre level.

► We also adjusted for a number of family level factors known to be related to children's anthropometry.

► We did not have a measure of each child's birth weight to account for children's individual growth trajectories, nor previous childcare attendance to assess prospective associations.

## INTRODUCTION

In 2010, 43 million preschool-aged children worldwide were estimated to be overweight or obese, with a further 92 million at risk of overweight/obesity.[1] Although levels appear to be stabilising,[2] obesity in childhood is associated with a range of unfavourable outcomes including type 2 diabetes, hyperlipidaemia and psychosocial problems.[3]

Obesity is often described as an imbalance between energy intake (food consumption) and energy expenditure (physical activity) resulting in excess weight gain over time.[4] As early childhood is a period of rapid growth and development, it represents a key time for establishing healthy energy-balance related behaviours (EBRBs), which include physical activity, and sedentary and dietary behaviours. While parents provide the majority of care for children before they enter school, children now spend increasing amounts of time in non-parental care prior to starting formal education.[5]

In 2014, the average enrolment rate of children aged 3–5 years in (preschool) educational programmes across Organisation for Economic Co-operation and Development countries was 80%.[5] In the UK, children aged 3–4 years have been entitled to 15 hours of free childcare (38 weeks per year) since 2010,[6] regardless of parental employment, and as of 2017, children aged 3–4 years of working parents may be eligible for up to 30 free hours per week.[7] Consequently, in 2017, 95% of UK children aged 3–4 years were enrolled in formal care,[8] attending for 21.7 hours per week on average.[9] Preschool-aged children,

**BMJ**

and particularly those who spend large amounts of time in formal care each week, therefore rely on childcare centres to provide a significant proportion of their food and physical activity opportunities. Elements such as staff education and training, staff behaviour on the playground, lower playground density (less children per square metre), the presence of vegetation and portable toys, and open play areas appear to be positively related to children's physical activity.[10] Likewise, childcare centres and specifically, individual childcare provider practices, appear to be associated with positive dietary behaviours in young children.[11]

Given the influence of the childcare environment on children's EBRBs, and the importance of EBRBs for children maintaining a healthy weight, it follows that time spent in childcare is likely to influence children's weight status. Indeed a number of studies assessing children during infancy and preschool suggest that children attending formal care (vs those not in formal care) are more likely to be overweight/obese,[12 13] with both nutrition and physical activity environments contributing.[14] Recently, three systematic reviews[15–17] have been conducted to evaluate the link between childcare attendance and obesity-related outcomes, assessing a combined total of 31 papers from North America (n=19), Europe (n=9) and Asia (n=3). Despite the heterogeneous nature of childcare definitions, childcare attendance appears to be associated with increased risk of overweight/obesity,[15–17] with informal care (ie, family member or non-relative) most commonly found to be associated with increased weight.[15 17] Several studies did however report a decreased risk or no association between informal or centre-based childcare and children's overweight/obesity.[16 17] In general, higher intensity childcare, especially when children have attended before 1 year, increased the overweight/obesity risk, with later (≥3 years old) enrolment in centre-based care associated with lower risk of overweight/obesity compared with earlier enrolment.[17] Lastly, there is some uncertainty as to whether risk differs by sociodemographic factors: children from lower socioeconomic backgrounds who spend time in childcare have been shown to have both a greater[18] and lower[19] risk of developing obesity.

Due to the high numbers of children attending formal childcare settings, and therefore the ability to reach large numbers of children in these environments, intervention studies are often conducted in these settings to prevent, halt or reverse obesity during the preschool years. A review by Zhou and colleagues[20] identified 15 such studies: 6 (of 13) interventions with a dietary component, and 8 (of 12) interventions with physical activity component reported improvements in the target EBRB,[20] with 7 (47%) reporting subsequent improvements in children's adiposity. Few studies (4/15) assessed long-term efficacy or sustainability (5/15) and the authors also noted heterogeneity across study designs and the interventions delivered, which suggests effects may be limited to specific population subgroups.

Such findings perhaps reflect differences in policies and practices around nutrition and physical activity across childcare centres (eg, by level of deprivation[21]) and between countries, which likely influence the extent to which the childcare environment is associated with children's weight indicators. Moreover, uptake of childcare varies between countries,[5] and so too does the amount of time children spend in any form of care,[18] which may also influence the strength of the association between the childcare environment and obesity levels. Finally, it is possible that despite the influence of the childcare environment, parental or family level factors, which are often targeted simultaneously, may exert a strong or stronger influence on children's weight status.[22]

In England, over one in five children are overweight or obese by the age of 5 years.[23] Despite high levels of childcare attendance in the UK, there has been very little research to assess associations between the childcare environment and children's health outcomes. With the publication of the UK Government's Obesity Plan for Action in 2017, strategies to encourage positive health behaviours and weight in preschool-aged children are increasingly centred on the childcare environment.[24] It is therefore timely to determine associations between the UK childcare environment and children's anthropometric indices. In this exploratory study, we therefore sought to assess how the amount of time spent in childcare, and how the nutrition, physical activity and overall childcare environment are associated with anthropometric indicators (body mass index z-score (z-BMI); waist-to-height ratio (WHR); sum of skinfold thickness (SST)) in a sample of UK children aged 3–4 years, adjusting for a range of family level explanatory variables. We hypothesised that children attending childcare centres with more supportive physical activity and nutrition environments would have favourable (ie, lower) anthropometric indices compared with those attending centres with less supportive environments.

## MATERIALS AND METHODS
### Study design and recruitment
Data were from the 'Studying Physical Activity in preschool-aged Children and their Environment (SPACE) Study', a cross-sectional childcare-based observational study.[25] The STrengthening the Reporting of OBservational studies in Epidemiology protocol was followed in the conduct and dissemination of this observational study. Participants were not involved in the development of the study design, research questions or outcome measures, but results were disseminated to all participants and participating childcare centres. Recruitment and data collection took place during January–July 2013; detailed information is available elsewhere.[25] Briefly, a list of preschool (state-run education) and nursery (privately run) 'childcare centres' in Cambridge were obtained from the Ofsted government website[26] and stratified by type (preschool/nursery) and tertile of Index of Multiple Deprivation (IMD; an area-level measure of deprivation[27]). Preschools and nurseries,

but not home/family based children centres, were purposively recruited because funding, the built environment and care provided tend to differ by type.[25] Within these six strata, 88 childcare centres were approached at random and invited in writing to participate; 30 (34%) centre managers consented to participate (n=15 preschool and n=15 nursery).

The parents of all children aged 3–4 years (n=602) attending consenting centres were sent a study invitation pack, and requested to return written consent to the childcare centre. Eligible children: were aged 3 years or 4 years; were free from physical disability; attended the centre for at least 9 hours per week (to ensure children spent >50% of their government-paid allocation at that particular centre); and were registered to attend the childcare centre on the designated measurement day. At least five children per centre with valid written consent (by a parent/legal guardian) were required to ensure sufficient analytical power. Children provided verbal assent prior to measurement (n=247, 41%).

## Patient and public involvement

Participants were not involved in the development of the study design, research questions or outcome measures, but results were disseminated to all participants and participating childcare centres.

## Data collection
### Child anthropometry and demographic data

At each centre visit, one of three trained researchers recorded each child's sex; measured height to the nearest 0.1 cm using a Leicester stadiometer; and weight to the nearest 0.1 kg using Seca digital scales in light indoor clothes with shoes and socks removed. Measures of weight and height were conducted once as these are highly reproducible with limited variability.[28] Abdominal waist circumference was measured to the nearest 0.5 cm at the midpoint between the lower costal margin and the level of the anterior superior iliac crests. Up to three measures were taken (if there was a discrepancy of 1 cm in the first two measures) using a Seca non-stretch tape measure next to the child's skin. Subscapular and tricep skinfolds were measured on the child's right side using a Holtain calliper (Holtain, UK) according to standard protocol.[29] If the first two measurements at either site were >0.2 mm apart, a third measurement was taken and an average calculated. Compared with a 'gold standard' trainer, researcher mean differences in measurement was 0.1 cm and 0.25–0.3 mm for waist and skinfolds, respectively. Equipment was calibrated prior to commencing data collection, at the midway point and on completion of the study.

Following anthropometric measurement, each child was allocated a study pack containing a parental questionnaire, which care providers disseminated to parents. The parental questionnaire, based on a previously validated questionnaire,[30] assessed demographic factors relating to the study participant; their general health and common

health behaviours; childcare attendance; other children in the home; family sociodemographics; parental occupational and leisure physical activity; parental height and weight; and parental beliefs, barriers and attitudes towards physical activity and nutrition. Parents were asked to return the questionnaire to their child's childcare centre 1 week later.

## Assessment of childcare environment

A trained researcher assessed the physical activity and nutrition environment of each centre using the Environment and Policy Assessment and Observation (EPAO) instrument. Observation began when the first child arrived in the morning and continued until the end of the day, when the last child left,[31] as the EPAO protocol requires a minimum of one full day to be spent observing all activities in the designated childcare classroom. Scoring is composed of two eight-item subscales for physical activity and nutrition. The physical activity subscales comprise: active opportunities, sedentary opportunities, sedentary environment, portable play environment, fixed play environment, staff physical activity behaviours, physical activity training and education, and physical activity policy. Nutrition subscales comprise: servings of fruits and vegetables; whole grains; high-sugar/high-fat foods; beverages; staff nutrition behaviours; nutrition environment; nutrition training and education; and nutrition policy.

As this tool was developed in the USA, a number of small amendments to the protocol and data collection template were made to ensure the tool was suitable in the UK context. This was done in consultation with the original development team to ensure the measure appropriately captured the physical activity and nutrition environments in UK childcare centres. These amendments did not change how the EPAO subscales for physical activity were scored; where preschools did not serve food, we used averages of available variables to derive the nutrition subscale scores.

## Variable derivation
### Outcome measures

Objectively measured height and weight were used to calculate children's BMI ($weight(kg)/ height^2(m)$). This, combined with child's sex and age in months at measurement (calculated from parental reported date of child's birth), was used to derive a continuous z-BMI, based on the British 1990 (UK90) growth reference charts.[32] International Obesity Task Force classifications were used to categorise children as normal weight, overweight or obese.[33] WHR (waist (cm)/height (cm)) was derived to assess central obesity, and the sum of subscapular and tricep skinfold thicknesses (SST) was derived as an indicator of subcutaneous fat.[34]

### Exposure variables

Parents were asked to provide information about their child's usual weekly childcare attendance using a

specifically designed question[25]: 'In a usual week, when does your child attend childcare? *Please only include care for your child taking part in SPACE and include regular formal and informal care (eg, grandparents, friends etc.)'*. Parents responded using free text, which we subsequently coded to derive the total number of reported hours children attended *formal* childcare during a usual week for analyses. Indicators of the overall childcare, nutrition and physical activity environments using EPAO (subscale) Scores were also generated. According to standard EPAO scoring procedure, responses to questions across eight physical activity, and up to eight nutrition subdomains were summed to a possible maximum of 20 points per domain (where each question was worth 0–2 points). The 'Physical Activity' Domain Score was derived using an average of the eight subscale domain scores for all centres. The 'Nutrition' Domain Score was calculated using an average of six **or** eight subdomain scores, depending on meals served. Many UK childcare centres do not serve all meals (ie, some only serve snacks/require children to bring packed lunch and do not provide lunch and 'tea' (at ~16:00 hours)). Two nutrition subscale scores (ie, Whole grains; High-sugar/high-fat foods) could therefore not be calculated for 21 centres: for these an average of the relevant six subdomain scores was calculated. An overall 'EPAO score' was derived by averaging the eight physical activity, and six or eight nutrition subdomain scores.

### Additional confounding variables

A range of potential explanatory variables relating to the child's family environment were derived using data collected from a parental questionnaire, including child ethnicity (White British; White European; other); maternal educational attainment (General Certificate of Secondary Education; A Levels; National Vocational Qualification/Diploma; University degree; Higher degree); maternal self-report BMI; and hours per week mothers worked (not employed; <20 hours; 21–35 hours; >35 hours). Where available, the latter three variables were also derived for fathers (with hours per week fathers worked categorised based on distribution as: <40 hours; 40–42 hours; >42 hours).

### Statistical analyses

All analyses were conducted using STATA V.14/SE. We calculated descriptive statistics for included participants, and compared them to those excluded from analyses (ie, those without valid anthropometric data) using independent t-tests and Pearson's $\chi^2$ test.

For each of our three continuous outcome variables (z-BMI, WHR, SST), a series of two-level mixed-effects linear regression analyses (level 1: child; level 2: childcare centre) were conducted to explore how weekly childcare attendance and practices relating to energy-balance behaviours in the childcare environment were related to children's anthropometric indices. Given children's age in months and sex are already taken into account

when z-BMI is derived, all WHR and SST analyses were adjusted for these variables. As children in the SPACE Study were recruited or 'clustered' at the centre level, multilevel regression analyses were used to allow for both within-centre and between-centre variations in anthropometric outcomes.[25 35] First, univariate analyses were conducted to assess how (A) weekly childcare attendance (in hours) and (B) the childcare environment relating to nutrition and physical activity, were associated with each of the three anthropometric indices. Multivariable analyses were then run, adjusting for family level confounding variables. One centre contributed two classes, which were treated as separate centres in analyses: although the classes shared policy documents, each ran as a completely independent entity, with different staff, rooms, children and outside spaces, and did not share catering facilities (hot meals were not provided).

Significant within-centre differences across the three outcome measures were identified. For example, 96% of the variation in z-BMI was explained by within-centre (ie, child-level) differences and only 4% of variation was explained by between-centre (ie, childcare-level) differences (within-centre variation WHR: 73%; STT: 85%). To further explore these differences, post hoc analyses were conducted to determine whether child-level socioeconomic status (measured using maternal educational attainment) moderated the association between time spent in the childcare environment and each outcome. Based on previous evidence, we hypothesised that the relationship between a more favourable childcare environment and normal weight would be stronger (ie, the gradient would be steeper) for children in lower-income families. This was grounded on the assumption that favourable childcare environments would buffer the association between potentially negative home environments and overweight/obesity in children from poorer socioeconomic backgrounds, but that the childcare environment may be less important for children from higher-income homes. Also, as fewer children provided SST data than z-BMI and WHR, sensitivity analyses were conducted to limit our analyses to children providing complete case-valid data for all three outcomes (n=151).

## RESULTS
### Participant characteristics

Of the 247 children who assented to have anthropometric measurements taken, valid anthropometric, observational and questionnaire data were available for 196 children from 30 centres (table 1). Childcare centres have been described previously[36]; those centres who participated did not differ in terms of area-level socioeconomic characteristics from those who declined to participate.[25] Children who provided complete case data (n=196) did not differ from those excluded by child's sex, age, weight status or ethnicity but were more likely to have mothers with higher education (higher degree: 38.3% vs 22.2%, P=0.02).

## Table 1 Descriptive characteristics of children included in analyses (SPACE Study, 2013, n=196)

| | Boys | Girls |
|---|---|---|
| **Child characteristics** | | |
| N (%) | 100 (51) | 96 (49) |
| Age (in months) | 47.4 (5.2) | 47.8 (4.9) |
| Ethnicity (N(%)) | | |
| White British | 74 (74.0) | 74 (77.1) |
| White other | 9 (9.1) | 9 (9.4) |
| Other/mixed ethnicity | 17 (17.0) | 13 (13.5) |
| z-BMI | 0.52 (0.93) | 0.36 (1.06) |
| Weight category* (N(%)) | | |
| Normal | 82 (82.0) | 79 (82.3) |
| Overweight/obese | 18 (18.0) | 17 (17.7) |
| Waist/height ratio | 0.49 (0.1) | 0.49 (0.1) |
| Centrally obese (N(%)) | 35 (46.7) | 28 (40.6) |
| Sum of skinfolds† (cm) | 14.5 (3.1) | 16.0 (3.6) |
| Average weekly hours in childcare | 23.3 (12.2) | 21.8 (12.1) |
| **Maternal characteristics** | | |
| Age (in years) | 37.6 (5.0) | 37.4 (5.5) |
| BMI (in kg/m$^2$) | 23.9 (4.2) | 24.2 (4.7) |
| Education (N(%)) | | |
| GCSE/A levels | 25 (25.0) | 34 (35.4) |
| Degree | 29 (29.0) | 33 (34.4) |
| Higher degree | 46 (46.0) | 29 (30.2) |
| Hours worked per week‡ (N(%)) | | |
| Not employed | 26 (27.4) | 24 (25.8) |
| <20 hours | 19 (20.0) | 14 (15.1) |
| 21–35 hours | 28 (29.5) | 31 (33.3) |
| >35 hours | 22 (23.2) | 24 (25.8) |
| **Paternal characteristics§** | | |
| Age (in years) | 39.2 (5.6) | 39.4 (8.0) |
| BMI (in kg/m$^2$) | 25.2 (3.5) | 25.9 (3.2) |
| Paternal education (N(%)) | | |
| GCSE/A levels | 23 (23) | 20 (24.1) |
| Degree | 24 (24) | 27 (32.5) |
| Higher degree | 43 (43) | 36 (43.4) |
| Hours worked per week‡ (N(%)) | | |
| <40 hours | 26 (29.8) | 30 (35.3) |
| 40–42 hours | 30 (35.6) | 30 (36.6) |
| >42 hours | 30 (34.5) | 22 (26.8) |

All values mean (SD) unless stated otherwise.
*Weight category derived using the International Task Force on Obesity cut points.
†n=80 boys and n=71 girls.
‡Categorised based on distribution, maternal employment: n=95 for boys, n=93 for girls.
§Paternal variables available for n=169–173 children depending on variable.
A levels, advanced levels; GCSE, General Certificate of Secondary Education; SPACE, Studying Physical Activity in preschool-aged Children and their Environment; z-BMI, body mass index z-score.

### Childcare environment

Domain subscale and total scores for EPAO are shown in table 2. Across childcare centres, the mean total EPAO Score (including eight physical activity and up to eight nutrition domains) was 11.2 (SD 1.0, range 8.5–13.5), with higher scores signifying more supportive environments. The average physical activity subscale score was 10.8 (1.5, 7.4–13.8) and average nutrition subscale score was 11.7 (1.6, 8.8–14.3). Overall, nutrition scores indicated good provision of fruit and vegetables, limited servings of high-fat/high-sugar foods, but poor provision of wholegrains. For physical activity, centres generally scored well for active opportunities and on staff physical activity behaviours; staff training and education in physical activity, and physical activity policies, were largely lacking.

### Association between childcare environment and children's anthropometric indices

In unadjusted analyses, there was no significant association between the number of hours a child spent in care or energy-balance childcare practices and preschool-aged children's anthropometric indices (table 3). These findings remained unchanged after adjusting for family level variables; as expected, several family level variables were independently associated with the outcomes of interest in adjusted models (see online supplementary tables S1–S3).

We found no significant interactions in post hoc analyses, designed to determine whether child-level socioeconomic status (measured using maternal educational attainment) moderated the association between time spent in the childcare environment and our three outcome measures (data not shown). In sensitivity analyses, including only children who provided all three outcome measures (n=151) did not significantly influence our findings (data not shown).

### DISCUSSION

We found that childcare attendance and energy-balance practices in the childcare environment do not appear to be associated with UK preschool-aged children's anthropometric indices. However, as shown previously, family level factors were independently associated with children's z-BMI. Children spend increasing amounts of time in formal childcare in the UK and the childcare environment is frequently the focus of intervention efforts to prevent or reduce early childhood obesity worldwide.[24] Childcare centres in the UK adhere to a statutory Early Years Foundation Stage (EYFS) framework, operate 'free-flow' policies where children generally choose their own activities with few provider-led activities, and by law must ensure all food and drink provided is properly prepared, wholesome and nutritious.[37] Therefore, this relatively standardised level of care may mean that UK childcare environments exert a smaller influence on children's EBRBs and health. Together, this suggests that child-level,

**Table 2** EPAO average domain subscale and total scores in the SPACE Study

| Nutrition subdomains* | Mean (SD) | Range | Physical activity subdomains† | Mean (SD) | Range |
|---|---|---|---|---|---|
| Servings: fruits and vegetables | 13.4 (4.1) | 6.7–20.0 | Active opportunities | 15.3 (3.8) | 6.7–20.0 |
| Servings: wholegrains‡ | 3.3 (3.3) | 0–10.0 | Sedentary opportunities | 11.7 (4.5) | 6.7–20.0 |
| Servings: high-sugar/high-fat foods§¶ | 14.5 (0.9) | 14.0–16.0 | Sedentary environment | 12.7 (4.1) | 6.7–20.0 |
| Servings: beverages | 9.7 (2.4) | 5.0–16.3 | Portable play environment | 10.7 (3.1) | 5.7–17.1 |
| Staff nutrition behaviours | 10.1 (2.8) | 3.3–15.0 | Fixed play environment | 11.8 (2.8) | 7.5–16.3 |
| Nutrition environment | 15.4 (3.0) | 6.7–20.0 | Staff PA behaviours | 14.3 (2.5) | 8.0–20.0 |
| Nutrition training and education | 10.6 (3.1) | 4.0–18.0 | PA training and education | 4.8 (4.0) | 0–15.0 |
| Nutrition policy | 12.6 (6.2) | 6.7–20.0 | PA policy | 7.0 (6.8) | 0–20.0 |
| Total Domain Score† | | | | | |
| Average Nutrition Domain Score** | 11.7 (1.6) | 8.8–14.3 | Average Physical Activity Domain Score†† | 10.8 (1.5) | 7.4–13.8 |
| Total EPAO Score† | | | | | |
| Average Total EPAO Score‡‡ | 11.2 (1.0) | 8.5–13.5 | | | |

*n=30 unless stated.
†n=30.
‡Centres n=9.
§Centres n=8.
¶Reverse coded such that higher score means favourable or lower provision of high-sugar, high-fat foods.
**Average of up to 8 subdomain scores.
††Includes 8 subdomain scores.
‡‡Average of up to eight nutrition and eight physical activity domain scores.
EPAO, Environment and Policy Assessment and Observation tool; PA, physical activity; SPACE, Studying Physical Activity in preschool-aged Children and their Environment.

family level, environmental-level and policy-level factors warrant significant further attention in obesity prevention strategies for young children.

### How this work compares to previous studies
Cross-sectionally, we did not identify any associations between time spent in care and children's anthropometry. This is in accordance with previous studies conducted in low-income Latino children aged 4 years[38] and children aged 2–5 years in Australia.[39] Interestingly, the latter study identified maternal education as the sole significant predictor of the child's weight status.[39] Although consideration of family level factors do not impact the conclusions drawn here (ie, no association was found before or after adjustment), several were independently associated with z-BMI, significantly attenuating the (non-significant) relationship between exposure and outcome. This confirms previous findings that family level factors may influence children's weight status, regardless of childcare attendance.[22]

Within the SPACE Study, there was relatively wide heterogeneity in design and layout of childcare centres. Yet EPAO Scores in this study showed comparatively little variation across childcare centres and were similar to those seen previously in US,[40] Dutch[36] and Canadian[41]

studies which suggests there were no obvious differences between these UK and other childcare environments. However, across and between countries differences may exist in how 'healthy' childcare environments are, and there may also be variation across physical activity and nutrition environments within the same childcare environment (ie, good nutrition, poor physical activity and vice versa). Taken together, the definition of a 'healthy' environment is likely to be relatively heterogeneous, potentially preventing identification of consistent associations.

It could be hypothesised that childcare in earlier childhood (ie, infancy), rather than during the preschool period (as assessed here), is more important for the development of children's adiposity. Indeed, work conducted in prospective cohorts suggests that early childcare attendance may be associated with children's weight status: Black and colleagues noted that early informal care (before 3 years) was associated with increased risk of overweight/obesity, and that higher childcare attendance, especially starting before 1 year, also increased overweight/obesity risk.[17] This was confirmed in studies in younger children (aged 1–2 years) from Ireland and the Netherlands. The latter suggests that attending childcare

**Table 3** Associations between childcare environment and preschool-aged children's anthropometric indices in the SPACE Study

| | Exposure measures β (95% CI) | | | | | | | |
|---|---|---|---|---|---|---|---|---|
| | Weekly hours in care | | EPAO PA Score | | EPAO Nutrition Score | | EPAO Total Score | |
| Continuous outcomes | Unadjusted | Adjusted* | Unadjusted | Adjusted* | Unadjusted | Adjusted* | Unadjusted | Adjusted* |
| z-BMI† | −0.00 (−0.02 to 0.01) | −0.01 (−0.02 to 0.00) | −0.02 (−0.12 to 0.09) | −0.01 (−0.12 to 0.11) | 0.01 (−0.09 to 0.10) | −0.01 (−0.11 to 0.09) | −0.00 (−0.01 to 0.01) | −0.00 (−0.01 to 0.01) |
| Waist to height ratio†‡ | −0.00 (−0.00 to 0.00) | −0.00 (−0.00 to 0.00) | 0.01 (−0.01 to 0.02) | 0.00 (−0.01 to 0.02) | −0.00 (−0.01 to 0.01) | −0.00 (−0.01 to 0.01) | 0.00 (−0.00 to 0.00) | 0.00 (−0.00 to 0.00) |
| Sum of skinfolds‡§ | 0.02 (−0.02 to 0.07) | 0.03 (−0.01 to 0.08) | 0.03 (−0.44 to 0.50) | 0.18 (−0.30 to 0.67) | −0.03 (−0.45 to 0.40) | −0.18 (−0.61 to 0.26) | 0.00 (−0.05 to 0.05) | 0.00 (−0.05 to 0.05) |

*Model adjusted for child ethnicity, maternal BMI, maternal educational attainment and maternal working hours.
†n=196.
‡Model also includes child sex and age in months.
§n=144.
EPAO, Environment and Policy Assessment and Observation tool; PA, physical activity; SPACE, Studying Physical Activity in preschool-aged Children and their Environment; z-BMI, body mass index z-score.

part-time or full-time was associated with increases in anthropometric indicators and odds of being overweight compared with those not in childcare prospectively.[42 43] Moreover, cumulative exposure to centre-based care up to the age of 4 years appears to be associated with higher odds of overweight/obesity in later childhood, that is, at 4–10 years of age,[12] in Canadian children. As such, these longitudinal associations appear to be dependent on the timing and level of exposure to childcare.[15 17] Greater childcare attendance in very early childhood appears to be associated with childhood obesity, but later childcare attendance does not show such associations.[17] As no data on previous childcare attendance patterns were collected in this study, we were unable to test this hypothesis here.

It is conceivable that bias or residual confounding, in addition to our sample size, limited our ability to find associations; it is also possible that a lack of power (ie, low β) resulted in a type 2 error, or us failing to find an association where one existed. This said, we were able to detect relatively small associations between the outcome and family level factors (eg, maternal BMI: β coefficient=0.05), and had suitable heterogeneity in our exposure measures to do so. Previous studies have identified associations with a range of time-based exposures assessed over longer periods of time (ie, 10 hours/ week increments[44]; each additional 30 days exposure to care[13] during the first year of life) or simply by the type of care children were exposed to (ie, childcare vs informal vs parental care[12 18]).[15] This variation in exposure and outcome measures is therefore also likely to account for differing findings between childcare-related factors and children's anthropometric indices.

This is, to our knowledge, one of the few studies to go beyond exploration of the association between time spent in care and anthropometry, to assess how elements within the childcare environment (ie, nutrition and physical activity practices) are related to children's anthropometric indices. In a previous study using this same sample of children, we found a similar lack of association between the childcare environment and preschool-aged children's physical activity[36] as identified here between the preschool environment and children's anthropometry. This is in contrast to studies in the USA[40] and Canada[45] suggesting that more favourable physical activity environments are associated with higher physical activity levels in children. It is possible that differences identified here are due in part to a fairly standardised system of care in the early years in the UK mentioned earlier. The EYFS framework[46] is a curriculum against which children's development is judged at age 2 years and 5 years (after their first year in primary school). A key tenant of this is that children must 'be helped to understand the importance of physical activity, and to make healthy choices in relation to food.' (page 5[46]). All food and drink served to children in UK early years' settings must by law be properly prepared, wholesome and nutritious, with fresh drinking water available to children at all times.[37] UK childcare centres additionally operate a 'free-flow' policy where children

may choose from a range of inside and outside activities for the majority of the day, regardless of weather conditions. With a standardised level of provision, the childcare environment in the UK may exert only a small influence on children's EBRBs and health outcomes. Yet with a core mandate to promote healthy EBRBs, and with high levels of children attending childcare, childcare settings in the UK appear well placed to ensure all children receive the best start in life, irrespective of sociodemographic factors.

To date, much of the research in this area has focused on the formal childcare environment. Strategies to encourage positive health behaviours and weight in preschool-aged children are therefore increasingly centred on the childcare environment, particularly in the UK.[24] However, very little research has been conducted in the UK childcare setting to determine whether such interventions are necessary or likely to succeed. Indeed, this and previous research[36] suggests that the formal childcare environment in the UK does not appear to be associated with preschool-aged children's anthropometric and physical activity outcomes. Other types of care (eg, family childcare homes/childminder; grandparents) however, also deserve attention. For example, in Canada, preschool-aged children in full-day kindergarten (providing instructive programmes for preschoolers) accumulated significantly more moderate-to-vigorous physical activity (MVPA) than those in centres (providing developmental programmes for a range of ages) or home-based care,[45] whereas in another study, EPAO nutrition subscales in centre-based care were shown to be more supportive than those in home-based care.[41] Given UK preschool-aged children spent an average of 15 hours in informal care in 2017,[9] more research is needed into the potential influences of these informal environments. This said, areas for improvement within the formal childcare environment likely still exist, with a more targeted approach, focusing on specific areas within the childcare environment (eg, promoting healthy dietary provision,[21] improving physical activity policies and staff training), still potentially benefiting children's health behaviours.

Finally, the children's family and wider environments should not be overlooked. Although the family is suggested to be central to health promotion in young children, both more proximal family and external factors (eg, in the community and wider environment) combine to shape a child's health behaviours.[47] Indeed, parents and childcare providers often cite each other as important custodians of preschool-aged children's EBRBs, suggesting both should work synergistically to encourage positive health and habit formation.[48]

### Strengths and limitations

We used objective measures of UK preschool-aged children's anthropometric indices to provide novel information about how attendance and the childcare environment are associated with children's anthropometric indices. Although this was a relatively small UK childcare-based sample, both our outcomes and exposures were normally distributed, and provided us with sufficient heterogeneity to explore our research questions. Although derivation of one of the exposure measures (ie, the Nutrition Domain Score) differed from previous studies, limiting the number of subscale scores used (to 6 rather than taking an average of up to 8) did not influence our findings. As one trained observer conducted the EPAO ratings, it is possible that bias or inaccuracy in these ratings may have occurred; the impact of this on the EPAO predictor variables is unknown. By using hierarchical regression analysis, we were also able to take account of study design and potential clustering at the centre level, thus increasing our power to detect (small) significant differences in our outcomes. We also adjusted for a number of family level factors known to be related to children's anthropometry but we did not have a measure of each child's birth weight to account for children's individual growth trajectories. As with several studies in this area,[15] this work was based on a childcare-based sample. We were therefore unable to assess the influence of no or only informal care, in addition to centre-based care, on children's weight-related outcomes.

Children were drawn from childcare centres recruited from the top tertile of IMD Scores in England.[25] As there were no differences in IMD Scores of the centres that did and did not participate in the SPACE Study,[36] children included here are likely representative of the wider eligible population. Children's individual socioeconomic circumstances (ie, maternal educational attainment) varied within this sample, which is important as children from both lower and higher socioeconomic backgrounds may be at risk of higher BMI.[18 19 23] It is therefore not clear how generalisable our findings are. However, we did not see evidence of a moderating effect of children's socioeconomic circumstances on the exposure-outcome relationship, suggesting that the association may be consistent across socioeconomic strata. Just under a fifth (18%) of children were classified as overweight or obese, which is slightly below the national UK average for 5-year-old children (22.5% in 2013/2014).[23] Children were predominantly White (British/European) (85%), which is in line with the UK average (86% in 2011).[49] Work is however required to determine whether similar findings are apparent in differing minority populations across the UK.

### CONCLUSIONS

We found no significant association between childcare attendance, or the nutrition and physical activity childcare environment, and children's anthropometric outcomes, suggesting the UK childcare environment had little influence on children's weight status in our study. In contrast, family level factors were associated. Although childcare-based interventions are increasingly the focus for promoting healthy weight and EBRBs in preschool-aged children, they tend to show small effects that are not sustained over time. Looking to other areas of a child's life, specifically family level factors and those in a child's

wider environment, either as an adjunct or alternative to centre-based interventions, should become a focus. Considering how a broader range of potential influences may interact to contribute to children's health will be key in successfully promoting healthy weight in preschool-aged children.

**Acknowledgements** The authors thank all children and their parents who participated in the SPACE Study. The authors also thank members of the MRC Epidemiology Unit for their assistance: members of the field team who conducted data collection, Kate Westgate and Stefanie Hollidge from the physical activity technical team for the accelerometer data processing, and Dr Emanuella De Lucia Rolfe and Richard Powell for processing the anthropometric data.

**Contributors** KRH and EMFvS conceived of the SPACE Study. KRH collected the data, conducted the analyses and drafted the manuscript. KRH, SEB-N and EMFvS interpreted the results and critically revised the manuscript. All authors approved the final manuscript and agree to be accountable for all aspects of the work.

**Funding** This work was conducted by the Medical Research Council (Unit Programme number MC_UU_12015/7) and the Centre for Diet and Activity Research (CEDAR), a UKCRC Public Health Research Centre of Excellence. The British Heart Foundation, Cancer Research UK, Economic and Social Research Council, Medical Research Council, the National Institute for Health Research, and the Wellcome Trust, under the auspices of the UK Clinical Research Collaboration, provided funding (CEDAR grant numbers: ES/G007462/1; 087636/Z/08/Z; MR/K023187/1). KRH is funded by the Wellcome Trust (107337/Z/15/Z). SEB-N is funded by NIH and the Robert Wood Johnson Foundation.

**Competing interests** All authors have completed the ICMJE uniform disclosure form at http://www.icmje.org/coi_disclosure.pdf (available on request from the corresponding author) and declare that (1) they have support from a number of grants outlined above; (2) all authors have no relationships with any organisations that might have an interest in the submitted work in the previous 3 years; (3) their spouses, partners or children have no financial relationships that may be relevant to the submitted work and (4) have no non-financial interests that may be relevant to the submitted work. Hesketh - UK Childcare environment and preschool-aged children's anthropometry.

**Patient consent** Not required.

**Ethics approval** University of Cambridge Psychology Ethics Committee (Pre.2012.68).

**Provenance and peer review** Not commissioned; externally peer reviewed.

**Data sharing statement** Unpublished data regarding children's health behaviours, socioeconomic circumstances and parental behaviours are available for the SPACE Study. The anonymised data set and statistical code used for analyses are available from the corresponding author on request. Model parameters are outlined in the 'Methods' section.

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
