## [Reviewer comments · BMJ Open]

ARTICLE DETAILS

TITLE (PROVISIONAL)	How does the UK childcare energy-balance environment influence 3-to-4-year-olds' anthropometry? A cross-sectional exploration
AUTHORS	Hesketh, Kathryn R.; Benjamin Neelon, Sara E.; van Sluijs, Esther

VERSION 1 – REVIEW

REVIEWER	Dr. Leigh Vanderloo The Hospital for Sick Children, Toronto, Canada
REVIEW RETURNED	20-Jan-2018

GENERAL COMMENTS	GENERAL COMMENTS Thank you for the opportunity to review this paper. The focus of this study is quite timely given the growing focus on the childcare environment as a health-promoting setting for young children, coupled with the large proportion of young children enrolled in childcare. I have included some suggestions that I believe will strengthen the overall quality and contribution of this work. Be consistent with your use of “preschool-aged child” vs. “preschooler”. I would choose one and be consistent throughout. What percentiles and/or cut-offs were used to define overweight and obesity among young children? Instead of “out-of-home care”, I would use the term “non-parental care”. Many children attend home- or family-based childcare which is run out of the childcare provider’s personal home. I think this would also help coincide with more of the international literature on this topic. I think the rationale for this paper could be stronger, or at the very least, more poignant. Why specifically did you choose to examine the relationship between the physical activity / nutrition childcare environment and obesity? In the methods, can you provide some additional information re: the recruitment and consent process. Were multiple classrooms from each centre participate? Did all children from each of the participating classrooms/centre receive consent to participate? What happened if a parent did not consent to their child participating? Analyses section – if multiple classroom from one centre participated, how was this handled in the analyses? Results – can the authors provide additional interpretation to the total domain scores for PA and nutrition? Given the average scores,
--

	I assume the environments – both the physical activity and nutrition – were not very supportive (due to low scores). Much research in Canada has been undertaken in the childcare environment using the EPAO. I encourage the authors to include additional work from international researchers. The take-home messages of the paper could be reinforced and better highlighted. I think the “so what?” piece is missing. If home/family-based childcare centres were included in this study (i.e., what was considered “informal centres”), the use of the EPAO tool with this environment needs to be reflected in the limitations sections as this tool was originally validated for the centre-based childcare environment. Please ensure references meet BMJ Open’s guidelines. Please review the reference list and revise accordingly. Please be consistent with your use of the Oxford comma throughout. SPECIFIC COMMENTS  - Introduction (page 3, line 48) – replace “whilst” with “while” - Introduction (page 4, line 4) – define OECD - Introduction (page 4, line 41) – by informal, do you mean family-and/or home-based care? - Methods (page 6) – indicate that the STROBE protocol was followed in this study - Methods (page 8) – can you explain what the range of the total EPAO score means? Higher scores are indicative of what? - Discussion (page 16) – I believe the sub-heading of this section should be “Discussion” as a second “conclusion” heading follows further on - Strengths and limitations (page 19, line 55) – correct the “what how” typo Respectfully submitted.
--	---

REVIEWER	Cathy Stough University of Cincinnati, USA
REVIEW RETURNED	25-Jan-2018

GENERAL COMMENTS	BMJ Review
	The manuscript “The childcare energy-balance environment and UK 3-4-year-olds’ anthropometry” addresses the important topic of the possible influence of childcare on child risk for obesity. The authors are commended for their use of objective measures of anthropometry, particularly in a community setting where this type of methodology can be difficult and is often lacking in other research. The manuscript is also exceptionally well-written and provides a thorough presentation of both the rationale for the study and interpretation of study findings. Overall Throughout the paper, the authors should pay attention to defining

all abbreviations the first time they are used. There are many instances of abbreviations not spelled out (e.g., z-BMI in the abstract, OECD, GCSEs, NVQ).

Abstract

The authors note (line 29) that “children provided valid data”. The measures were completed either by parents or taken by research assistants, so I think this could be better worded. Something like: Valid data were available for 196 children.

The authors draw the conclusion that family-level factors (line 4) warrant substantial attention in future research. While I certainly agree, I think this statement excludes the potential of other factors (e.g., community, child-level) that may also be important for future attention. The current study suggests that childcare factors may not be appropriate for future focus, but the goal was not to assess what other specific factors might be important. While it can be drawn from the control variables in the analyses that family factors are important, this seems to be a “side” finding from the study, since identifying whether family factors was important was not an actual focus. Therefore, I think it is equally important to also mention the importance of assessing the potential influence of other factors beyond the school environment. This comment is also relevant to the Conclusions section, where only family-level factors are mentioned as a possible influence on child obesity risk.

Introduction

Minor Change: Typo on Page 4 Line 32: “systematic reviews” should be singular.

Page 5 Line 5: The authors comment that there is ease in intervening in childcare environments. I do not necessarily see this as the case, since trying to make changes in a setting that involves many stakeholders and policies can actually be quite difficult. I would remove this reasoning or provide additional support or justification for why making changes in childcare environments is easy.

Page 5 Line 46: The authors present information regarding obesity rates in England. Is there any data regarding out-of-home childcare usage in England? More general numbers for multiple countries are presented on page 4, but details of childcare usage in England specifically would make it clear how many children the findings of this manuscript may generalize to.

Materials and Methods

Page 6 Line 21: Were there any inclusion criteria for centres? The authors mentioned that more detailed information is available elsewhere, so I do not think you need to go into detail here, but a brief sentence would be helpful.

Study Design and Recruitment: The authors do not include any

information about child assent or parental consent to study procedures. I would like to see something added regarding when parent informed consent was obtained and how parents were informed about the study. How did parents have the option to decline? E.g., was this study conducted with passive consent or did parents actually complete informed consent? There is a brief mention of this in the data sharing statement, but no details in the manuscript.

Data Collection: No information is provided about how parent-report data were collected. Please add information about how the parent questionnaire was completed and more details about this questionnaire.

Page 6 Line 38: Were height and weight only measured once? The authors note that other anthropometrics were measured multiple times, but it is not mentioned whether this was the case for height and weight. If multiple measurements were not completed, why? If multiple measures were collected, which values were used in computation of BMI?

Page 6 Line 53: Please cite the standard protocol used for collecting skinfold measurements.

Page 7 Line 12: Was any reliability calculated for EPAO observations/ratings? For example, did multiple researchers complete ratings for a subset of centres in order to calculate reliability of observations?

Page 7 Line 51: I am having a little trouble understanding how nutrition subscale scores were computed for schools that did not serve meals. The authors mention that averages of available variables were used. Does this mean other Nutrition variables or is this referring to the Physical activity variables? If meals were not served, does that mean all Nutrition variables were missing or were some still able to be observed?

Results

Did the authors explore whether specific Physical Activity or Nutrition subscales related to anthropometric outcomes? I imagine it could be possible that specific subscales are related to anthropometrics even if the overall total score is not. I know your power to do this is probably lacking and since this would include so many comparisons some correction for Type 1 error would be necessary. If this was looked at and no significant findings emerged, I think it is fine to just not include any of this.

Conclusions

Related to my comment under the abstract section, the authors make note of the importance of examining family-level factors in future research, but I do not think we can assume this is necessarily the most important future direction as other factors (e.g., child,

	community) may also be important. Page 18 Lines 35- 43: These statements seem a little contradictory to me. The first sentence seems to say that research has focused on interventions to encourage positive health behaviors in child care settings, but then the second sentence says very little intervention research has been conducted. Strengths and Limitations: There are a couple of either typos or difficult to read phrases in this section (Line 38 “the influence of no or only”, Line 55 “what how”). Tables I am having trouble understanding what the multiple “Unadjusted” and “Adjusted” columns in supplemental table 1 represent. Why is there not just 1 unadjusted column and 1 adjusted column? How do all the “unadjusted” columns differ from each other (and same for adjusted)? Using bold font to note the significant relationships might help ease reading the tables.
--	---

REVIEWER	fionnuala mcauliffe university college Dublin ireland
REVIEW RETURNED	29-Jan-2018

GENERAL COMMENTS	This is a study of 196 3-4 year olds from 30 childcare centres The main findings were that time spent in care, and the nutrition, physical activity, and overall childcare environment were not associated with children’s zBMI score, WHR and SST. This study adds to the existing literature on this important topic of the inter-relation between childcare and childhood adiposity Other studies have found relationships between childcare, especially informal childcare and childhood adiposity and it would be useful to reference them eg Child Care Exposure Influences Childhood Adiposity at 2 Years: Analysis from the ROLO Study. Scully H, Alberdi G, Segurado R, McNamara A, Lindsay K, Horan M, Hennessy E, Gibney E, McAuliffe F. Child Obes. 2017 Apr;13(2):93-101 Exploration of different setting of childhood would strengthen the discussion and possible strategies that could be used to target families would be useful
---

VERSION 1 – AUTHOR RESPONSE

Reviewers' Comments to Author:

Reviewer: 1

Reviewer Name: Dr. Leigh Vanderloo

Institution and Country: The Hospital for Sick Children, Toronto, Canada

Competing Interests: None declared.

GENERAL COMMENTS

Thank you for the opportunity to review this paper. The focus of this study is quite timely given the growing focus on the childcare environment as a health-promoting setting for young children, coupled with the large proportion of young children enrolled in childcare. I have included some suggestions that I believe will strengthen the overall quality and contribution of this work.

Thank you for your positive comments – we are pleased you agree this is a timely piece of work.

Be consistent with your use of “preschool-aged child” vs. “preschooler”. I would choose one and be consistent throughout.

We appreciate you pointing out this inconsistency. We have now used the term preschool-aged child/ren throughout.

What percentiles and/or cut-offs were used to define overweight and obesity among young children?

We have now included the following:

The International Obesity Task Force classifications to categorise children as normal, overweight or obese.[21]

Instead of “out-of-home care”, I would use the term “non-parental care”. Many children attend home- or family-based childcare which is run out of the childcare provider’s personal home. I think this would also help coincide with more of the international literature on this topic.

Thank you for making this very valid point, we have now changed this as suggested.

I think the rationale for this paper could be stronger, or at the very least, more poignant. Why specifically did you choose to examine the relationship between the physical activity / nutrition childcare environment and obesity?

We have now included the following in the introduction:

In England, over 1 in 5 children are overweight or obese by the age of five.[23] Despite high levels of childcare attendance in the United Kingdom (UK), there has been very little research to assess associations between the childcare environment and children’s health outcomes. With the publication of the UK Government’s Obesity Plan for Action in 2017, strategies to encourage positive health behaviours and weight in preschool-aged children are increasingly centred on the childcare environment.[24] It is therefore timely to determine associations between the UK childcare environment and children’s anthropometric indices. In this exploratory study, we therefore sought to

assess how the amount of time spent in childcare, and how the nutrition, physical activity and overall childcare environment are associated with anthropometric indicators (z-BMI score; waist-to-height ratio (WHR); sum of skinfold thickness (SST)) in a sample of UK 3-to-4-year-old children, adjusting for a range of family-level explanatory variables. We hypothesised that children attending childcare centres with more supportive physical activity and nutrition environments would have favourable anthropometric indices compared to those attending centres with less supportive environments.

In the methods, can you provide some additional information re: the recruitment and consent process. Were multiple classrooms from each centre participate? Did all children from each of the participating classrooms/centre receive consent to participate? What happened if a parent did not consent to their child participating?

In response to your queries, we have now included the follow in the methods section:

The parents of all children aged 3-4 years (n=602) attending consenting centres were sent a study invitation pack, and requested to return written consent to the childcare centre. Eligible children were: aged 3 or 4 years; free from physical disability; attended the centre for at least nine hours per week (to ensure children spent >50 % of their government-paid allocation at that particular centre); and registered to attend the childcare centre on the designated measurement day. At least five children per centre with valid written consent (by a parent/legal guardian) were required to ensure sufficient analytical power. Children provided verbal assent prior to measurement. The University of Cambridge Psychology Ethics Committee provided ethical approval for the study (Pre.2012.68).

Analyses section – if multiple classroom from one centre participated, how was this handled in the analyses?

There was only one instance where one centre contributed two classes with 3-4-year-old children. In this case, they were treated as separate settings. This was because although the classes shared centre policy documents, each class ran as completely independent entities. Each had different staff, rooms, children and outside spaces. The centre also only facilitated packed lunches so classes did not share any catering facilities or snack preparation. We have now included the following in the Methods:

One centre contributed two classes, which were treated as separate centres in analyses: although the classes shared policy documents, each ran as a completely independent entity, with different staff, rooms, children and outside spaces, and did not share catering facilities (hot meals were not provided).

Results – can the authors provide additional interpretation to the total domain scores for PA and nutrition? Given the average scores, I assume the environments – both the physical activity and nutrition – were not very supportive (due to low scores).

We have now included the following in the results section:

Domain subscale and total scores for the EPAO are shown in Table 2. Across childcare centres, the mean total EPAO score (including eight physical activity and up to eight nutrition domains) was 11.2 (SD: 1.0, range: 8.5 - 13.5), with higher scores signifying more supportive environments. The average physical activity subscale score was 10.8 (1.5, 7.4-13.8) and average nutrition subscale score was 11.7 (1.6, 8.8-14.3). Overall, nutrition scores indicated good provision of fruit and vegetables, limited

servings of high-fat, high-sugar foods, but servings of wholegrains were poor. For physical activity, centres generally scored well for active opportunities and staff physical activity behaviours, but staff training and education in physical activity, and physical activity policies, were largely lacking.

And refer to how these values compare to previous international studies in the Discussion:

EPAO scores in this study were similar to those seen previously in US,[38] Dutch,[35] and Canadian[39] studies which suggests there were no obvious differences between these UK and other childcare environments.

Much research in Canada has been undertaken in the childcare environment using the EPAO. I encourage the authors to include additional work from international researchers.

We have now included a revised summary of the childcare and weight status literature in the introduction (see pages 4/5) and have also included a number of additional references to international work in the Discussion (please refer to pages 20-23).

The take-home messages of the paper could be reinforced and better highlighted. I think the “so what?” piece is missing.

Thank you for highlighting this. We agree that the implications of these findings could be better elucidated and we have included the following in the discussion:

We found that childcare attendance and energy-balance practices in the childcare environment do not appear to be associated with UK preschool-aged children’s anthropometric indices. However, as shown previously, family-level factors were independently associated with children’s z-BMI score. Children spend increasing amounts of time in formal childcare in the UK and the childcare environment is frequently the focus of intervention efforts to prevent or reduce early childhood obesity worldwide.[24] Childcare centres in the UK adhere to a statutory Early Years Foundation Stage (EYFS) framework, operate ‘free-flow’ policies where children generally choose their own activities with few provider-led activities, and by law must ensure all food and drink provided is properly prepared, wholesome and nutritious.[36] The UK childcare environment may therefore exert a smaller influence on children’s EBRBs and health. Together, this suggests that child-, family-, environmental-, and policy-level factors warrant significant further attention in obesity prevention strategies for young children.

If home/family-based childcare centres were included in this study (i.e., what was considered “informal centres”), the use of the EPAO tool with this environment needs to be reflected in the limitations sections as this tool was originally validated for the centre-based childcare environment.

As we only included preschool or nursery settings in the SPACE study, no home/family-based childcare centres were included here. We have now clarified the methods section as follows:

Preschools and nurseries, but not home/ family-based children centres, were purposively recruited as funding, the built environment and care provided tend to differ by type.[20]

Please ensure references meet BMJ Open's guidelines. Please review the reference list and revise accordingly.

We have now revised the references to comply with BMJ Open's guidelines.

Please be consistent with your use of the Oxford comma throughout.

We have amended the paper to ensure consistency.

SPECIFIC COMMENTS

- Introduction (page 3, line 48) – replace “whilst” with “while”

Amended

- Introduction (page 4, line 4) – define OECD

Amended

- Introduction (page 4, line 41) – by informal, do you mean family- and/or home-based care?

By informal care, we were referring to the definition given by Black and colleagues, but realise that this wasn't clear. We have also slightly revised this paragraph to incorporate more of the recent review evidence available. This now reads:

Despite the heterogeneous nature of childcare definitions, childcare attendance appears to be associated with greater overweight/obesity,[13–15] with informal care (relative or non-relative) most commonly found to be associated with increased weight.[13,15]

- Methods (page 6) – indicate that the STROBE protocol was followed in this study

Amended thanks:

The STrengthening the Reporting of OBservational studies in Epidemiology (STROBE) protocol was followed in the conduct and dissemination of this observational study.

- Methods (page 8) – can you explain what the range of the total EPAO score means? Higher scores are indicative of what?

Thank you for pointing out this omission – we have now clarified that higher scores mean more supportive environments.

- Discussion (page 16) – I believe the sub-heading of this section should be “Discussion” as a second “conclusion” heading follows further on

Amended

- Strengths and limitations (page 19, line 55) – correct the “what how” typo

Amended

Reviewer: 2

Reviewer Name: Cathy Stough

Institution and Country: University of Cincinnati, USA

Competing Interests: None declared.

The manuscript “The childcare energy-balance environment and UK 3-4-year-olds’ anthropometry” addresses the important topic of the possible influence of childcare on child risk for obesity. The authors are commended for their use of objective measures of anthropometry, particularly in a community setting where this type of methodology can be difficult and is often lacking in other research. The manuscript is also exceptionally well-written and provides a thorough presentation of both the rationale for the study and interpretation of study findings. Thank you very much for your kind comments.

Overall

Throughout the paper, the authors should pay attention to defining all abbreviations the first time they are used. There are many instances of abbreviations not spelled out (e.g., z-BMI in the abstract, OECD, GCSEs, NVQ).

Thank you for pointing this out. We have now defined all abbreviations when they first occur.

Abstract

The authors note (line 29) that “children provided valid data”. The measures were completed either by parents or taken by research assistants, so I think this could be better worded. Something like: Valid data were available for 196 children.

We have now amended this to:

Valid data were available for 196 children (49% female).

The authors draw the conclusion that family-level factors (line 4) warrant substantial attention in future research. While I certainly agree, I think this statement excludes the potential of other factors (e.g., community, child-level) that may also be important for future attention. The current study suggests that childcare factors may not be appropriate for future focus, but the goal was not to assess what other specific factors might be important. While it can be drawn from the control variables in the analyses that family factors are important, this seems to be a “side” finding from the study, since identifying whether family factors was important was not an actual focus. Therefore, I think it is equally important to also mention the importance of assessing the potential influence of other factors beyond the school environment. This comment is also relevant to the Conclusions section, where only family-level factors are mentioned as a possible influence on child obesity risk.

Thank you for raising this very valid point. We have now included the following in the abstract:

The childcare environment has been central to intervention efforts to prevent/ reduce early childhood obesity, yet other factors, including child-, family-, wider environmental, and policy-level factors warrant substantial attention when considering obesity prevention strategies for young children.

The discussion of the main manuscript:

We found that childcare attendance and energy-balance practices in the childcare environment do not appear to be associated with UK preschool-aged children's anthropometric indices. However, as shown previously, family-level factors were independently associated with children's z-BMI score. Children spend increasing amounts of time in formal childcare in the UK and the childcare environment is frequently the focus of intervention efforts to prevent or reduce early childhood obesity worldwide.[24] Childcare centres in the UK adhere to a statutory Early Years Foundation Stage (EYFS) framework, operate 'free-flow' policies where children generally choose their own activities with few provider-led activities, and by law must ensure all food and drink provided is properly prepared, wholesome and nutritious.[36] Therefore, this relatively standardised level of care may mean that UK childcare environments exert a smaller influence on children's EBRBs and health. Together, this suggests that child-, family-, environmental-, and policy-level factors warrant significant further attention in obesity prevention strategies for young children.

And in the conclusion of the main manuscript:

Looking to other areas of a child's life, specifically family-level factors and those in a child's wider environment, either as an adjunct or alternative to centre-based interventions, should become a focus. Considering how a broader range of potential influences may interact to contribute to children's health will be key in successfully promoting healthy weight in preschool-aged children.

Introduction Minor Change: Typo on Page 4 Line 32: "systematic reviews" should be singular.

Amended

Page 5 Line 5: The authors comment that there is ease in intervening in childcare environments. I do not necessarily see this as the case, since trying to make changes in a setting that involves many stakeholders and policies can actually be quite difficult. I would remove this reasoning or provide additional support or justification for why making changes in childcare environments is easy.

We have now amended this to:

Due to the high numbers of children attending formal childcare, and therefore the ability to reach large numbers of children in these environments, intervention studies are often conducted in these settings to prevent, halt or reverse obesity during the preschool years.

Page 5 Line 46: The authors present information regarding obesity rates in England. Is there any data regarding out-of-home childcare usage in England? More general numbers for multiple countries are presented on page 4, but details of childcare usage in England specifically would make it clear how many children the findings of this manuscript may generalize to.

Thank you for pointing out this omission – we have now included the following in the introduction:

In the UK, 3-to-4-year-old children have been entitled to 15 hours of free childcare (38 weeks per year) since 2010,[6] regardless of parental employment, and as of 2017, 3-to-4-year-olds of working parents may be eligible for up to 30 free hours per week.[7] Consequently, in 2017, 95% of UK 3-to-4-year-olds were enrolled in formal care,[8] attending for 21.7 hours per week on average.[9]

Materials and Methods

Page 6 Line 21: Were there any inclusion criteria for centres? The authors mentioned that more detailed information is available elsewhere, so I do not think you need to go into detail here, but a brief sentence would be helpful.

We have now included the following:

Briefly, a list of preschool (state-run education) and nursery (privately-run) 'childcare centres' in Cambridge were obtained from the Ofsted government website[21] and stratified by type (preschool/nursery) and tertile of index of multiple deprivation (IMD; an area-level measure of deprivation[22]). Preschools and nurseries, but not home/ family-based children centres, were purposively recruited as funding, the built environment and care provided tend to differ by type.[20] Within these six strata, 88 childcare settings were approached at random and invited to participate in writing; 30 (34%) centre managers consented to participate.

Study Design and Recruitment: The authors do not include any information about child assent or parental consent to study procedures. I would like to see something added regarding when parent informed consent was obtained and how parents were informed about the study. How did parents have the option to decline? E.g., was this study conducted with passive consent or did parents actually complete informed consent? There is a brief mention of this in the data sharing statement, but no details in the manuscript.

Thank you for highlighting this oversight. We have now included the following:

The parents of all children aged 3-4 years (n=602) attending consenting centres were sent a study invitation pack, and requested to return written consent to the childcare centre. Eligible children were: aged 3 or 4 years; free from physical disability; attended the centre for at least nine hours per week (to ensure children spent >50 % of their government-paid allocation at that particular centre); and registered to attend the childcare centre on the designated measurement day. At least five children per centre with valid written consent (by a parent/legal guardian) were required to ensure sufficient analytical power. Children provided verbal assent prior to measurement.

Data Collection: No information is provided about how parent-report data were collected. Please add information about how the parent questionnaire was completed and more details about this questionnaire.

We have now added the following to the amended 'Child Anthropometry and Demographic data' section:

Following anthropometric measurement, each child was allocated a study pack containing a parental questionnaire, which care providers disseminated to parents. The parental questionnaire, based on the previously validated questionnaire,[29] assessed demographic factors relating to the study

participant; their general health and common health behaviours; childcare attendance; other children in the home; family socio-demographics; parental occupational and leisure physical activity; parental height and weight; and parental beliefs, barriers and attitudes towards physical activity and nutrition. Parents were asked to return the questionnaire to their child's childcare centre one week later.

Page 6 Line 38: Were height and weight only measured once? The authors note that other anthropometrics were measured multiple times, but it is not mentioned whether this was the case for height and weight. If multiple measurements were not completed, why? If multiple measures were collected, which values were used in computation of BMI?

Yes, height and weight were measured only once, as is standard protocol. This is because they are highly reproducible and measurement is less variable compared to skinfolds and waist measurements.

We have added the following to the paper:

At each centre visit, one of three trained researchers recorded each child's sex; measured height to the nearest 0.1cm using a Leicester stadiometer; and weight to the nearest 0.1kg using Seca digital scales in light indoor clothes with shoes and socks removed. Measures of weight and height were conducted once as these are highly reproducible with limited variability.

Page 6 Line 53: Please cite the standard protocol used for collecting skinfold measurements.

We have now added the following reference:

Subscapular and tricep skinfolds were measured on the child's right side using a Holtain caliper (Holtain Ltd, United Kingdom) according to standard protocol.[26] If the first two measurements at either site were >0.2mm apart, a third measurement was taken and an average calculated. Compared to a 'gold standard' trainer, researcher mean differences in measurement was 0.1cm and 0.25-0.3mm for waist and skinfolds respectively. Equipment was calibrated prior to commencing data collection, at the mid-way point and on completion of the study.

[26] <http://dapa-toolkit.mrc.ac.uk/anthropometry/objective-methods/simple-measures-skinfolds>

Page 7 Line 12: Was any reliability calculated for EPAO observations/ratings? For example, did multiple researchers complete ratings for a subset of centres in order to calculate reliability of observations?

One researcher conducted all EPAO observations, so no reliability measures were calculated. However, the researcher underwent training in the USA to conduct EPAO assessment, where she passed standard testing procedures.

Page 7 Line 51: I am having a little trouble understanding how nutrition subscale scores were computed for schools that did not serve meals. The authors mention that averages of available variables were used. Does this mean other Nutrition variables or is this referring to the Physical

activity variables? If meals were not served, does that mean all Nutrition variables were missing or were some still able to be observed?

Thank you for highlighting that this was not clear. For all settings, we were able to calculate 6 nutrition subscales (Servings of fruits and vegetables; Beverages; Staff nutrition behaviours; Nutrition environment; Nutrition training and education; and Nutrition policy). As some settings required children to bring packed lunches (instead of meals cooked on the premises), we were unable to calculate two subscales for these settings (i.e. wholegrains; high sugar/high fat foods). We therefore calculated a nutrition score using an average of the available subscale scores (either 6 or 8). For all centres, the physical activity score was calculated using an average of 8 subscale scores. The total EPAO score was again calculated as an average of the available subscale scores (i.e. 14 or 16 subscale scores).

We have now amended the paper as follows:

The 'Physical Activity' domain score was derived using an average of the eight subscale domain scores for all centres. The 'Nutrition' domain score was calculated using an average of six or eight subdomain scores, depending on meals served. Many UK childcare centres do not serve all meals (i.e. some only serve snacks/ require children to bring packed lunch, and do not provide lunch and 'tea' (at ~4pm)). Two nutrition subscale scores (i.e. Whole grains; High-sugar/high-fat foods) could therefore not be calculated for 21 centres: for these an average of the relevant six subdomain scores was calculated. An overall 'EPAO score' was derived by averaging the eight physical activity, and six or eight nutrition subdomain scores.

Results

Did the authors explore whether specific Physical Activity or Nutrition subscales related to anthropometric outcomes? I imagine it could be possible that specific subscales are related to anthropometrics even if the overall total score is not. I know your power to do this is probably lacking and since this would include so many comparisons some correction for Type 1 error would be necessary. If this was looked at and no significant findings emerged, I think it is fine to just not include any of this.

Thank you for raising this. Having conducted the analyses we present in the paper, we did explore whether each of the subdomain subscales were associated with anthropometric indices. There were no significant findings and as you note, we are likely underpowered to explore these analyses formally. We therefore chose not to raise this in the paper, and are pleased you agree that this does not need to be included in the paper.

Conclusions

Related to my comment under the abstract section, the authors make note of the importance of examining family-level factors in future research, but I do not think we can assume this is necessarily the most important future direction as other factors (e.g., child, community) may also be important.

Thank you. As in the abstract, we have now altered the conclusions to reflect a broader range of factors that might be important for children's anthropometric indices.

Page 18 Lines 35- 43: These statements seem a little contradictory to me. The first sentence seems to say that research has focused on interventions to encourage positive health behaviors in child care settings, but then the second sentence says very little intervention research has been conducted.

Apologies that this was not clear. We were trying to make the point that although focus is increasingly shifting towards the childcare environment and the need to intervene here, very little research has been conducted in the UK to show that interventions are a) necessary or b) likely to succeed. Indeed our work suggests that the childcare environment is not associated with children's weight/ physical activity and limited resources may be better focused elsewhere. We have now amended this section to:

To date, much of the research in this area has focused on the formal childcare environment. Strategies to encourage positive health behaviours and weight in preschool-aged children are therefore increasingly centred on the childcare environment, particularly in the UK.[22] However, very little research has been conducted in the UK childcare setting to determine whether such interventions are necessary or likely to succeed. Indeed, this and previous research[32] suggests that the UK childcare environment does not appear to be associated with preschool-aged children's anthropometric and physical activity outcomes.

Strengths and Limitations: There are a couple of either typos or difficult to read phrases in this section (Line 38 "the influence of no or only", Line 55 "what how").

These have now been corrected.

Tables

I am having trouble understanding what the multiple "Unadjusted" and "Adjusted" columns in supplemental table 1 represent. Why is there not just 1 unadjusted column and 1 adjusted column? How do all the "unadjusted" columns differ from each other (and same for adjusted)?

Using bold font to note the significant relationships might help ease reading the tables.

Apologies that this was confusing. As in Table 3 in the paper, each of the 4 unadjusted and adjusted columns relate to each of the four exposure measures used in analyses (Weekly hours in care; PA score; Nutrition score; EPAO total score). We have now amended the supplementary tables in line with Table 3 and hope that these are now easier to interpret.

Reviewer: 3

Reviewer Name: fionnuala mcauliffe

Institution and Country: university college Dublin ireland

Competing Interests: none declared

This is a study of 196 3-4 year olds from 30 childcare centres

The main findings were that time spent in care, and the nutrition, physical activity, and overall childcare environment were not associated with children's zBMI score, WHR and SST.

This study adds to the existing literature on this important topic of the inter-relation between childcare and childhood adiposity

We are very pleased that you think this is an important topic for study.

Other studies have found relationships between childcare, especially informal childcare and childhood adiposity and it would be useful to reference them eg

Child Care Exposure Influences Childhood Adiposity at 2 Years: Analysis from the ROLO Study.

Scully H, Alberdi G, Segurado R, McNamara A, Lindsay K, Horan M, Hennessy E, Gibney E, McAuliffe F. *Child Obes.* 2017 Apr;13(2):93-101

Thank you for highlighting this literature, which we have now included:

This was confirmed in studies in younger children (aged 1-2 years) from Ireland and The Netherlands. The latter suggest that attending childcare part- or full-time was associated with increases in anthropometric indicators and odds of being overweight compared to those not in childcare prospectively.[42,43]

Exploration of different setting of childhood would strengthen the discussion and possible strategies that could be used to target families would be useful

Thank you for this helpful suggestion. We have now included the following in the Discussion:

To date, much of the research in this area has focused on the formal childcare environment. Strategies to encourage positive health behaviours and weight in preschool-aged children are therefore increasingly centred on the childcare environment, particularly in the UK.[24] However, very little research has been conducted in the UK childcare setting to determine whether such interventions are necessary or likely to succeed. Indeed, this and previous research[35] suggests that the formal childcare environment in the UK does not appear to be associated with preschool-aged children's anthropometric and physical activity outcomes. Other types of care (e.g. family child care homes/ childminder; grandparents) however also deserve attention. For example in Canada, preschool-aged children in full day kindergarten (providing instructive programmes for preschoolers) accumulated significantly more MVPA than those in centres (providing developmental programmes for a range of ages) or home-base care,[44] whereas in another study, EPAO nutrition subscales in centre-based care were shown to be more supportive than those in home-based care.[39] Given UK preschool-aged children spent an average of 15 hours in informal care in 2017,[9] more research is needed into the potential influences of these informal environments. This said, areas for improvement within the formal childcare environment likely still exist, with a more targeted approach, focusing on specific areas within the childcare environment (e.g. promoting healthy dietary provision[21], improving physical activity policies and staff training), still potentially benefitting children's health behaviours.

Finally, the children's family and wider environments should not be overlooked. Although the family is suggested to be central to health promotion in young children, both more proximal family and external factors (e.g. in the community and wider environment) combine to shape a child's health behaviours.[47] Indeed, parents and childcare providers often cite each other as important custodians of preschool-aged children's EBRBs, suggesting both should work synergistically to encourage positive health and habit formation.[48]

VERSION 2 – REVIEW

REVIEWER	Dr. Leigh Vanderloo The Hospital for Sick Children, Toronto, Canada
REVIEW RETURNED	28-Mar-2018

GENERAL COMMENTS	I think the authors have done an extremely thorough job at addressing all of my noted comments/concerns. Well done!
---

REVIEWER	Cathy Stough University of Cincinnati, USA
REVIEW RETURNED	02-Apr-2018

GENERAL COMMENTS	Thank you to the authors for their comprehensive and attentive responses to feedback. I appreciate the thoroughness and responsiveness of changes to the manuscript and the response to reviewers letter. I have some very minor comments and suggestions. Introduction It is great that the authors have added some a priori hypotheses to the introduction. It would be even more helpful if the authors could be more specific regarding hypotheses. Specifically, the authors mention that “environments would have favourable anthropometric indices compared to those...” What is meant by “favourable” (could you say “lower”)? And listing out the anthropometric indices (perhaps just putting in parentheses following “anthropometric indices) would be helpful. I appreciate the authors adding more clarity about what is meant by “informal care” in the Introduction. However, I find that it is still a little difficult to understand what is meant by this term, largely due to my lack of familiarity with the Black et al definition. For example, does the term “relative” refer to “family member”? Methods The Methods section was improved by the authors’ addition of further details about recruitment. You state “the parents of all children aged 3-4 years (n = 602) attending”, but do not include in this section the number of participants who consented and subsequently what your participation rate was. It would be great to add here the number who consented and the % this was of the total number of people that were sent a study invitation pack. The sentence “preschools and nurseries, but not home/family-based children centres, were purposively recruited as funding, the built environment”... would be clearer if “as” was replaced with “because”. Thank you for adding information regarding how parent-report data was collected. In the sentence: “The parental questionnaire, based on the previously validated questionnaire” it is unclear which questionnaire you are referring to when you say “previously validated questionnaire”. I also appreciate the clarification that weight and height were only
---

	measured 1 time. This is different than the standards and conventions I am used to. Could you add a citation supporting that these measurements are “highly reproducible with limited variability”? It is certainly a limitation of the study that there was no reliability check conducted on EPAO ratings. Bias or inaccuracy in the observer’s ratings could still be possible even though they are trained, and the impact of potential bias and inaccuracy on the main predictor variables cannot be assessed. This should be added to the Discussion section as a limitation. Results The following sentences were a little difficult to follow given the grammar and commas. Is there a way to reword this more clearly?: “Overall, nutrition scores indicated good provision of fruit and vegetables, . . . and physical activity policies, were largely lacking”. Discussion In the sentence “Childcare centres in the UK adhere to a statutory EYFS framework, operate ‘free-flow’ policies were children”, the “were” should be “where”
--	--

VERSION 2 – AUTHOR RESPONSE

Introduction

It is great that the authors have added some a priori hypotheses to the introduction. It would be even more helpful if the authors could be more specific regarding hypotheses. Specifically, the authors mention that “environments would have favourable anthropometric indices compared to those...” What is meant by “favourable” (could you say “lower”? And listing out the anthropometric indices (perhaps just putting in parentheses following “anthropometric indices) would be helpful. We do not feel it is necessary to re-list the anthropometric indices as these appear three lines before in our paper aims. However, we have now clarified the direction of the proposed association as suggested:

We hypothesised that children attending childcare centres with more supportive physical activity and nutrition environments would have favourable (i.e. lower) anthropometric indices compared to those attending centres with less supportive environments.

I appreciate the authors adding more clarity about what is meant by “informal care” in the Introduction. However, I find that it is still a little difficult to understand what is meant by this term, largely due to my lack of familiarity with the Black et al definition. For example, does the term “relative” refer to “family member”?

We have now amended this to family member.

Methods

The Methods section was improved by the authors’ addition of further details about recruitment. You state “the parents of all children aged 3-4 years (n = 602) attending”, but do not include in this section the number of participants who consented and subsequently what your participation rate was. It would be great to add here the number who consented and the % this was of the total number of people that were sent a study invitation pack.

The number of children who assented to be measured is included in the Results section, but we have also now included this as % of those approached in the methods:

Children provided verbal assent prior to measurement (n=247, 41%).

The sentence “preschools and nurseries, but not home/family-based children centres, were purposively recruited as funding, the built environment”... would be clearer if “as” was replaced with “because”.

We have changed the ‘as’ for ‘because’ as suggested.

Thank you for adding information regarding how parent-report data was collected. In the sentence: “The parental questionnaire, based on the previously validated questionnaire” it is unclear which questionnaire you are referring to when you say “previously validated questionnaire”.

We have now amended this sentence to:

The parental questionnaire, based on a previously validated questionnaire,[30] assessed demographic factors relating to the study participant;

I also appreciate the clarification that weight and height were only measured 1 time. This is different than the standards and conventions I am used to. Could you add a citation supporting that these measurements are “highly reproducible with limited variability”?

We have now added in the following reference:

- [1] Crespi C, Alfonso A, Whaley S, et al. Validity of child anthropometric measurements in the Special Supplemental Nutrition Program for Women, Infants and Children. *Pediatr Res* 2012;**71**:286–92.

It is certainly a limitation of the study that there was no reliability check conducted on EPAO ratings. Bias or inaccuracy in the observer’s ratings could still be possible even though they are trained, and the impact of potential bias and inaccuracy on the main predictor variables cannot be assessed. This should be added to the Discussion section as a limitation.

We have now included the following in the Strengths and Limitations section:

As one trained observer conducted the EPAO ratings, it is possible that bias or inaccuracy in these ratings may have occurred; the impact of this on the EPAO predictor variables is unknown.

Results

The following sentences were a little difficult to follow given the grammar and commas. Is there a way to reword this more clearly?: “Overall, nutrition scores indicated good provision of fruit and vegetables, . . . and physical activity policies, were largely lacking”.

We have now revised this sentence to:

For physical activity, centres generally scored well for active opportunities and on staff physical activity behaviours; staff training and education in physical activity, and physical activity policies, were largely lacking.

Discussion

In the sentence “Childcare centres in the UK adhere to a statutory EYFS framework, operate ‘free-flow’ policies were children”, the “were” should be “where”

Thank you for noting this error – this has now been amended.

VERSION 3 – REVIEW

REVIEWER	Cathleen Stough University of Cincinnati, USA
REVIEW RETURNED	17-May-2018
GENERAL COMMENTS	The authors have addressed all previous comments well. I do not have any further recommendations for changes.